# Exploration of How Uncertainty Tolerance, Emotion Regulation, and Hope Are Linked and Influenced in People with Chronic Low Back Pain: A Worked Example of a Social Constructivist Meta-Ethnography Study

**DOI:** 10.3390/bs15101399

**Published:** 2025-10-15

**Authors:** Justine McMillan, Andrew Soundy

**Affiliations:** 1School of Sport, Exercise and Rehabilitation Sciences, University of Birmingham, Birmingham B15 2TT, UK; a.a.soundy@bham.ac.uk; 2North Devon District Hospital, Royal Devon University Healthcare NHS Foundation Trust, Raleigh Heights, Barnstaple, Devon EX31 4JB, UK

**Keywords:** chronic low back pain, hope, uncertainty, intolerance of uncertainty, emotion regulation, social constructivist meta-ethnography

## Abstract

Chronic low back pain (CLBP) is a significant global concern. Its prevalence continues to rise, and current management strategies demonstrate, at best, moderate effectiveness. The purpose of this study was to explore the concept of uncertainty tolerance and how it affects an individual’s ability to hope, as well as how both of these factors influence emotion regulation. These concepts are highly relevant for both patients and clinicians during CLBP consultations. A social constructivist meta-ethnographic study—a highly interpretative type of qualitative review—was conducted to generate new theory and provide valuable insights into this unique area of pain management. A framework was followed; its iterative analytical process involves multiple search strategies in accordance with PRISMA guidelines. The analysis explored relationships among studies, generated conceptual ideas, and ultimately developed a substantive theory. This review represents the first worked example of a theory generating review process. One model is created to illustrate how individuals with CLBP regulate their emotions; it also considers both the influencing factors and the resulting outcomes of this process. The model identifies pathways leading to either adaptive or maladaptive emotional regulation strategies. Tolerance of uncertainty and ability to hope are key concepts that influence emotion regulation and play a vital role in the physical and psychological well-being of people with CLBP. Further research is required to explore how the model can be operationalised in clinical practice.

## 1. Introduction

Chronic low back pain (CLBP) is one of the most prevalent musculoskeletal conditions globally, defined as pain lasting more than three months, extending beyond the typical recovery period ([28]; [73]). Its prevalence has steadily increased since the 1990s and is expected to continue rising, significantly affecting individuals’ physical and psychological well-being, as well as society ([22]). Current management strategies recommend a combination of physical and psychological approaches, but their effectiveness is moderate at best, with supporting evidence being of low quality ([32]; [36]; [56]; [57], [58]; [81]). Pain is a complex phenomenon involving neurophysiological and psychological components ([77]). In 2021, the International Association for the Study of Pain redefined pain as “an unpleasant sensory and emotional experience associated with, or resembling that associated with, actual or potential tissue damage” ([67]). Understanding pain requires exploring interrelated psycho-emotional constructs—uncertainty, emotion regulation, and hope—that are particularly relevant in CLBP. This article examines these constructs and their implications for theory and practice in this patient group.

Uncertainty is closely linked to CLBP due to its complexity ([27]). It is defined as a cognitive state arising from encountering “an unknown” or a perceived absence of information ([13]), which reduces one’s ability to predict outcomes and one’s sense of control, often triggered by inconsistent symptoms or unfamiliar events ([16]). Qualitative studies show that people with CLBP frequently feel uncertain about their diagnosis, symptom management, and the condition’s impact on their self-identity and future ([18]; [22]; [86]). Uncertainty affects emotional responses and regulation, influencing mental and physical well-being ([30]). Although emotion remains a debated concept, it is generally understood as a time-limited physiological state shaped by personal experience, appraisal, and behavioural expression ([13]; [30]). Several neural systems are involved in emotion activation, which varies depending on the stimulus and individual response ([7]). Emotional responses are especially important in patient–clinician interactions, where poor uncertainty tolerance can impair problem-solving and increase distress ([87]).

Intolerance of uncertainty is defined as “a dispositional inability to endure the discomfort caused by missing key information” ([13]), and can manifest cognitively (e.g., negative interpretation), emotionally (e.g., worry), and behaviourally (e.g., avoidance) ([17]; [87]). Initially linked to generalised anxiety disorder ([12]), it is now recognised as a transdiagnostic construct across psychological disorders ([70]). The intolerance of uncertainty model explains its association with worry through three mechanisms: positive beliefs about worry, negative problem orientation, and cognitive avoidance ([63]). However, the model does not fully explain individual responses to different types of uncertainty or which aspects are most difficult to manage. Responses to intolerance of uncertainty are shaped by prior experiences, including childhood development, attachment security, cognitive flexibility and sociocultural influences ([27]; [50]; [70]). Further development of models addressing uncertainty in CLBP interactions would be valuable, especially given the complexity of association between different interrelated constructs.

The ability to regulate emotions is essential for managing life’s challenges and significantly influences mental and physical well-being ([7]). Although emotion regulation has been widely studied in behavioural and neuroscience research ([7]; [37]; [73]), its application to chronic pain is relatively recent. Emotion regulation involves consciously or automatically adjusting one’s emotional state using emotion regulation strategies (ERSs) ([86]). These strategies are broadly classified as explicit (deliberate) or implicit (automatic) ([7]). Explicit regulation includes conscious efforts such as reappraisal, valuing the present, or choosing to hope ([79]). Implicit regulation operates unconsciously, including automatic responses to stimuli or goal-directed behaviours without awareness, e.g., making healthier choices or walking more by parking further away ([50]). Automatic responses may involve re-evaluating stimuli based on new experiences (e.g., a sound once linked to a negative event losing its impact over time when no longer associated with that event). Controlled implicit processes can guide behaviour toward goals without conscious intent. These nonconscious goals are particularly important during periods of stress (([50]). Ultimately, different ERSs are driven by distinct neural systems, resulting in varied behavioural outcomes ([7]), which is important to understand in healthcare professional–patient interactions.

Interactions between healthcare professionals and patients with CLBP are strongly influenced by perceptions of possibility or uncertainty. The ability to identify a positive future and maintain hope depends on an individual’s tolerance for uncertainty and emotional regulation. Healthcare professionals must be aware of these concepts, as they can negatively affect how patients view their future ([79]). Close attention is needed to understand how uncertainty and hope are managed in clinical interactions ([74]). Clinicians should recognise that establishing hope fosters optimism and positive emotions such as joy and anticipation, acting as a protective factor for mental health ([84]). Other protective factors include the use of cognitive strategies and cognitive flexibility (being able to adapt one’s thinking) to establish meaningful goals ([74]; [79]). Conversely, uncertainty can lead to a perception of hopelessness and negative emotions, resulting in poor psychological outcomes such as severe depression, and in some cases, distant thoughts or fantasies about death as an escape from persistent pain ([78]; [82]). This may be particularly important at times of onset, during exacerbations or periods of change, uncertainty about outcomes or when patients view their future as predetermined ([79]). Research on hope has primarily focused on non-chronic pain populations, such as cancer, ageing, and chronic illnesses like heart disease and multiple sclerosis, consistently showing that higher levels of hope correlate with greater pain tolerance, improved physical health, and psychological well-being ([35]; [61]). Only recently have uncertainty and hope begun to be explored in chronic pain populations, particularly in chronic musculoskeletal conditions ([22]; [61]). Hope plays a vital therapeutic role in healthcare by improving prognostic and behaviour-related outcomes ([47]) and serves as a key component of patients’ coping mechanisms ([78]). If healthcare professionals can effectively navigate uncertainty, foster possibility, and support patient hope during CLBP consultations, they may deliver more tailored and potentially therapeutic care ([74]).

There is a complex relationship between hope, uncertainty, and emotion regulation that requires a greater conceptual understanding in order to benefit interactions between healthcare professionals and patients with CLBP. Although research on emotion regulation in chronic pain is limited, it highlights the importance of both implicit and explicit regulation and its relevance to chronic pain, warranting further investigation ([37]; [66]). To date, no studies have explored how factors that influence uncertainty may be understood within a single model and process that related to interaction. Addressing this gap requires a theory-generating review. Social constructivist meta-ethnography is a new methodology designed to develop substantive theory ([75]). Therefore, this study aimed to explore uncertainty tolerance, its impact on hope, and how both influence emotion regulation in people with CLBP, using the social constructivist meta-ethnography framework.

## 2. Methods

The methodology used in this study was social constructivist meta-ethnography, a modified version of the traditional meta-ethnography framework. It incorporates phases from social constructivist grounded theory to ensure analytical generalisability and support critical enquiry ([66]). This is important because the traditional version of meta-ethnography often results in a poorly considered model or process as an output ([75]; [76]). In contrast, the social constructivist meta-ethnography uses iterative analytical processes and techniques originally derived from social constructivist grounded theory to ensure a substantive theory or process as an output. Moreover, in the later stages of the analytical process, the theory or model also undergoes rigorous testing and modifications until theoretical saturation is reached, ensuring rigour ([75]). As such, this methodology generates a theory that is co-created from the literature and, inevitably, the authors’ interpretations, biases, and experiences ([15]). However, the potential value is the ability to challenge current thinking and generate analytical generalisability ([11]). The reader is encouraged to use the framework ([75]) to understand the choices and phases in more detail. The researcher’s philosophical position is grounded in social constructivism,; a paradigm that combines a pragmatic ontological stance with a relativist epistemology ([76]).

### 2.1. Protocol and Registration

The protocol for this study was registered with the International Prospective Register of Systematic Reviews (PROSPERO) on 5 January 2024 (registration number CRD42024493925) and was updated throughout the process to reflect the inevitable changes to the research questions and eligibility criteria due to the nature of this methodology, as previously discussed.

### 2.2. Initial Eligibility Criteria

Studies were considered for inclusion by two blind reviewers using Covidence© (JM/AS). Studies were included if they used a sample of adults (aged over 18 years) with CLBP (low back pain persisting for more than three months). For studies that used a small sub-group of participants with acute LBP as a comparison (*n* = 3), a discussion was undertaken between reviewers to consider the study, the sample, its contribution to the model, and whether it should be included. All three studies were included. Additionally, for any study which included participants were under 18 years of age (*n* = 1), a similar process and discussion were undertaken. This study was included as only one participant was under 18 years of age. Only studies that were written in English were included, and no date restriction was applied. Lastly, studies were required either to have used an outcome that considered the phenomenon of hope or to have discussed the concept of hope from the perspective of the individual with CLBP in the results section. Studies that explored the experience of an intervention to improve hope were not included.

Due to the nature of the methodology, additional searching was required. This was due to the iterative processes involved in the methodology, and the need to challenge the development of a substantive theory and help ensure theoretical saturation. Please see Step 2 in Figure 1 (Section 2.7 Synthesis) for details on the process. In brief, two additional complete systematic searches were undertaken by the same two blind reviewers (JM/AS) to investigate the concepts of uncertainty, and then later on, emotion regulation (Step 5, Figure 1). The total search numbers for all additional searches described here are included in each PRISMA flow diagram (See Figure 2, Figure 3 and Figure 4).

### 2.3. Search Strategy for Qualitative Literature

The key requirement of meta-ethnography is to bring together qualitative studies on a particular topic. In the current review, a total of three systematic literature searches were undertaken blind by both authors and supported by the Covidence© software (Version 2) on 20 May 2025. All searches were designed to capture experiences of the three major concepts inputted into the model. The primary search associated with the initial eligibility criteria involved searches of MEDLINE, CINAHL Plus, AMED, ERIC, SPORTDiscus, and the Hope-Lit database. In addition, the first 10 pages or 100 articles on electronic search engines such as Google Scholar and ScienceDirect were screened. Grey literature was searched using the GreyMatters search engine. Standard Boolean operators were used. Keywords included hope, hopelessness, hope scale, chronic low back pain, non-specific low back pain, persistent low back pain, pain management, pain reduction, and quality of life, but excluded optimism, as this is a more general belief that things will work out for the best and is considered a different construct ([84]). The same databases and search engines were used for the next two systematic searches. Keywords for the second systematic search included uncertainty, uncertain, intolerance of uncertainty, possibility, and chronic low back pain, non-specific low back pain, and persistent low back pain. Lastly, keywords for the third systematic search included emotion regulation, emotion dysregulation, regulation of emotion, chronic low back pain, non-specific low back pain, and persistent low back pain. Further details of all systematic searches are outlined in the audit trail (see Appendix A).

### 2.4. Study Selection and Data Extraction Approach

Duplicates were identified via the Covidence software©. Articles were screened and selected independently by both authors by reading the title and abstract, followed by the full text. Conflicting decisions were resolved and justifications for study exclusions are provided in the audit trail (see Appendix A). Table 1, Table 2 and Table 3, in Section 3.1
*“Search Outputs”*, summarise the demographic details of the originally included empirical studies.

### 2.5. Quality of Included Articles

[75] ([75]) identifies that four principal questions should be considered regarding the included studies in order to meet the aims of critical enquiry. The questions are as follows: (a) Are considerations and information given by the selected articles made sufficiently well so that concepts can be translated? (b) Do findings provide a context for the culture, environment, and setting? (c) Are the findings relevant and useful given the focus or aims of the analysis now? (d) Do the questions asked or aims from the paper selected align with those sought by the meta-ethnographer? (e) To what extent do the findings give theoretical insight and context of interpretation made? The quality scores of the included articles can be found in Table 4 in Section 3.2
*“Quality Considerations”*.

### 2.6. Generalisability of Results and Searching for Conceptual Models That May Assist with Analytical Generalisability

The topic of generalisation within qualitative research is greatly debated, which is largely influenced by the researcher’s philosophical worldview ([11]). The focus of this study was to generate a substantive theory and achieve analytical generalisability by using a framework that draws on iterative phases of theory development ([75]). This type of generalisation draws conclusions from singular studies which are then used to develop a broader theory that is co-created with the main researcher’s interpretation, experiences, and biases ([40]). To enhance, expand, and challenge the model created, we identified studies which represented concepts that could help explain, influence, or represent an outcome from an uncertain or unknown situation. This resulted in the model being expressed and refined 10 times (see Appendix A for the 10 different versions of the model) as the identification and testing of different aspects and elements of the model took place. Based on the qualitative synthesis, the following concepts were explored, examined, and justified with regards to their inclusion for the main theory: cognitive flexibility, emotional regulation, BAS activation, predisposing factors (including patient history), and hope. This was to ensure that the literature already included in the review (containing some individuals with acute low back pain) could be examined according to the implications provided for the model. As part of illustrating a worked example, the entire analytical process is clearly outlined—with justifications—in the audit trail (see Appendix A) to enhance transparency for the reader.

### 2.7. Synthesis

An eight-step approach was used (see Figure 1), based on the social constructivist meta-ethnography framework ([75]). A detailed account of this process can be found in the audit trail (see Appendix A).

As outlined in Step 2 of the synthesis process (see Figure 1), the initial articles exploring hope were reviewed and coded, which highlighted a possible link to uncertainty. This theme was sub-categorised into diagnostic uncertainty, prognostic uncertainty, and the individual’s beliefs or perceptions about the unknown. These findings support Mishel’s theory ([51]) on uncertainty in healthcare, which states that, regardless of the underlying health condition, uncertainty arises when individuals cannot cognitively appraise information about the state of their illness—particularly if the course of the disease is unpredictable—or when there is a lack of information about the diagnosis and/or prognosis. Moreover, [79] ([79]) who developed a framework for hope, also recognised that hope is particularly challenged at the time of onset, during periods of change, or in the presence of uncertainty. Therefore, this represented a critical turning point in the analytical process, and the concept of uncertainty became a new line of enquiry and representing a phenomenon which was critical to the development of a substantive theory.

The eligibility criteria were expanded to include the discussion of uncertainty in the results section, and a second systematic search was undertaken for the same population. After immersive reading and coding of these articles, four key psychosocial themes emerged based on uncertainty, which were as follows: (1) the individual’s beliefs about the unknown, (2) the clinical encounter and diagnosis, (3) the impact on their self-identity, social relationships, and future, and (4) treatment failure. Subsequently, how patient–clinician interaction can influence these areas of uncertainty, either positively or negatively, was explored. Following this, it was important to establish intrinsic factors such as personality traits, and pre-existing conditions, such as emotional disorders or neurodevelopmental conditions, that can predispose an individual to be less tolerant of uncertainty and to explore how their ERS differs from those without such conditions. As a result, a third systematic search exploring the concept of emotion regulation within people with CLBP was undertaken. Figure 2, Figure 3 and Figure 4 presents the PRISMA flow diagrams that outlines the approach for each systematic search (further details of the search process can be found in Appendix A).

**Figure 1 behavsci-15-01399-f001:**
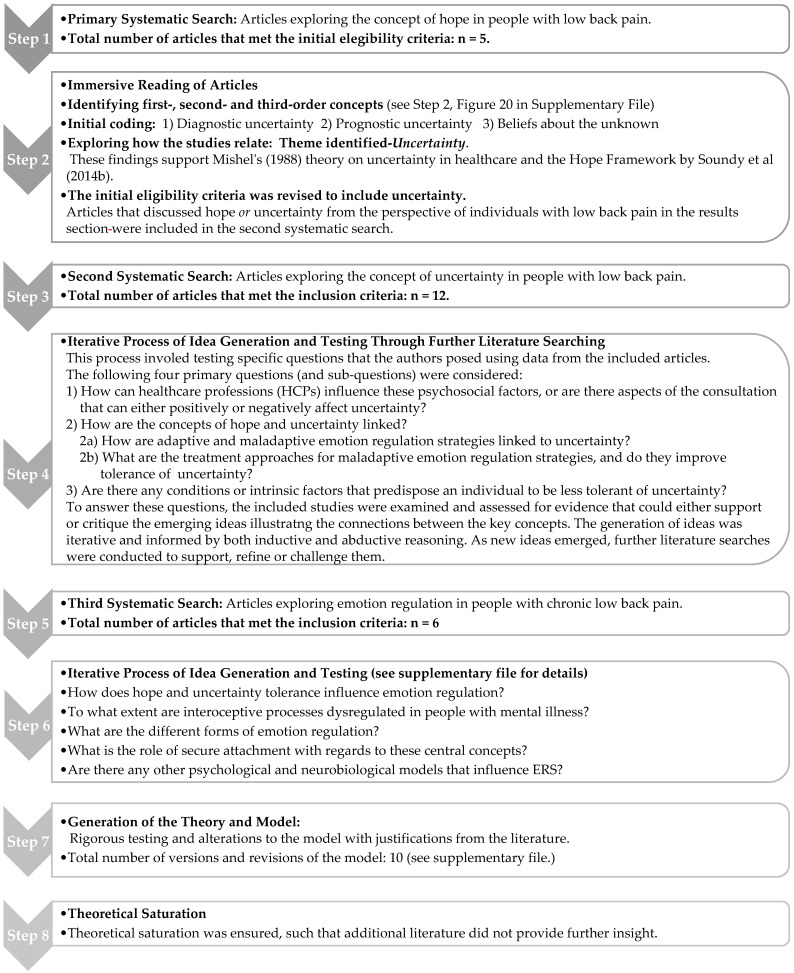
The synthesis process (search and concept output). [51]’s ([51]) and [79] ([79]).

**Figure 2 behavsci-15-01399-f002:**
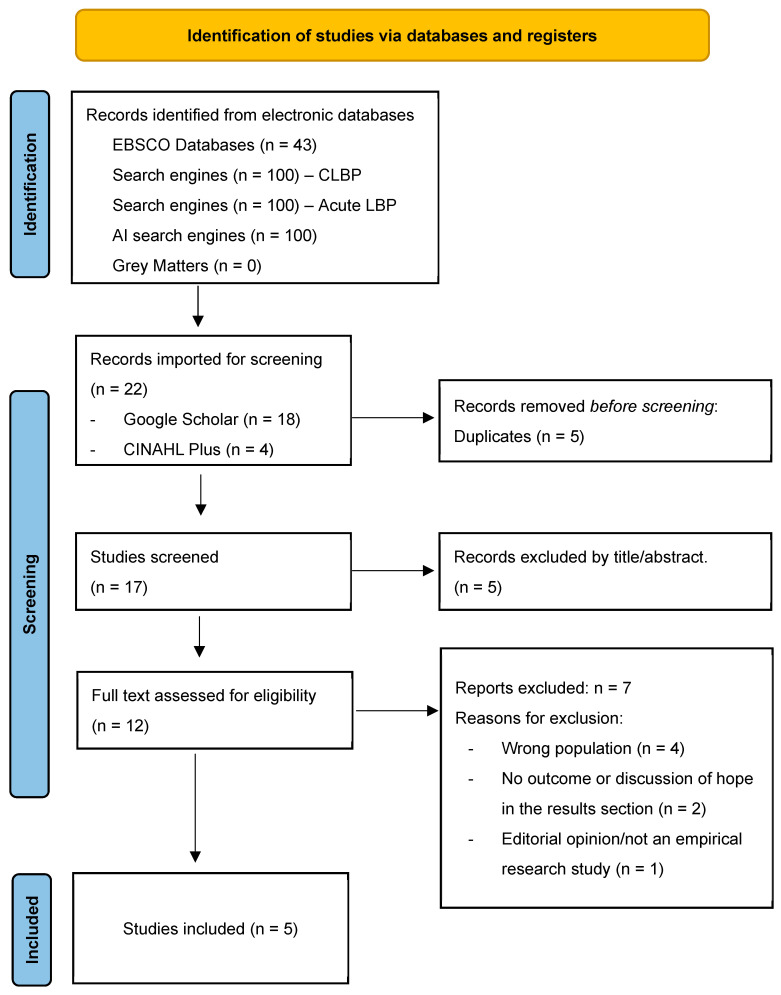
Presents the PRISMA flow diagram that outlines the results of the first systematic search for articles exploring the concept of hope in people with CLBP ([64]).

The eligibility criteria evolved, enabling two further searches to be reported. Figure 3 and Figure 4 present PRISMA flow diagrams outlining the process of the second and third systematic search, respectively.

**Figure 3 behavsci-15-01399-f003:**
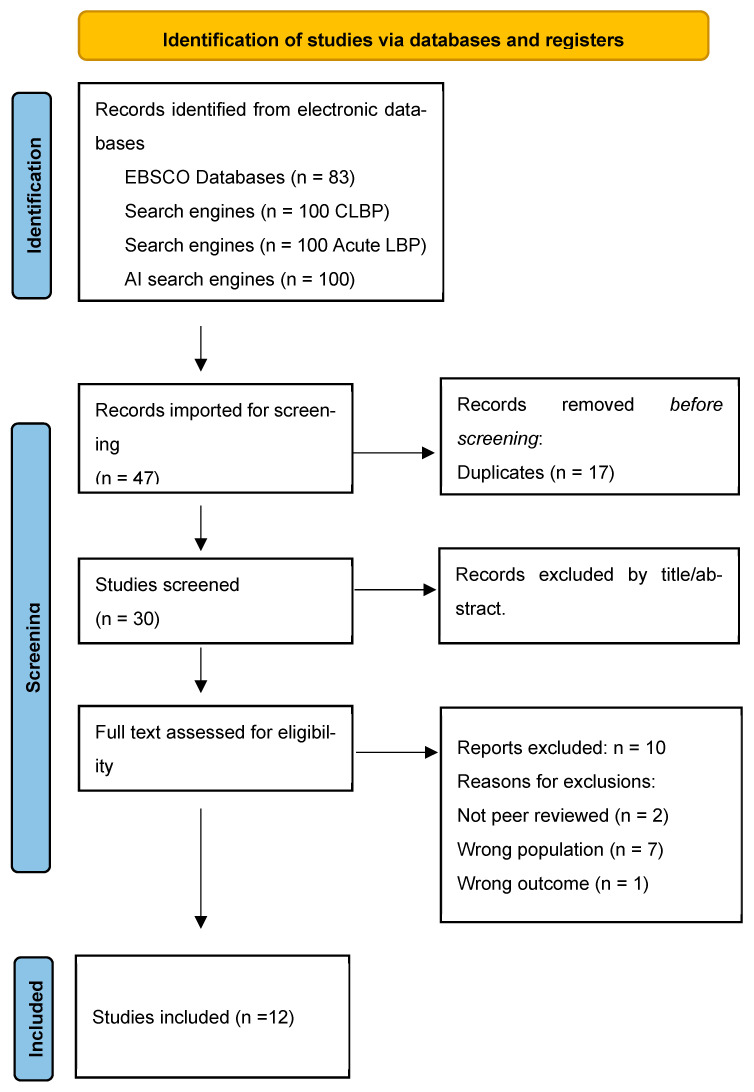
Presents the PRISMA flow diagram that outlines the results of the second systematic search for articles exploring the concept of uncertainty in people with CLBP ([64]).

**Figure 4 behavsci-15-01399-f004:**
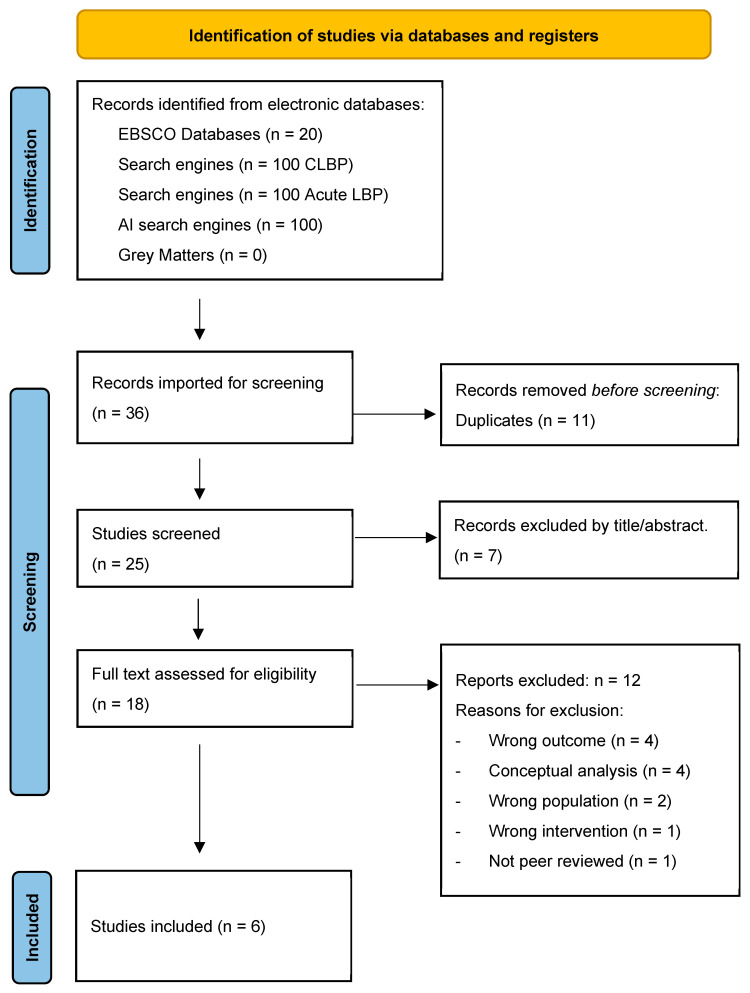
The PRISMA flow diagram that outlines the results of the third systematic search for articles exploring the concept of emotional regulation in people with CLBP ([64]).

## 3. Results

### 3.1. Search Outputs

Due to the inductive nature of this methodology, multiple searches are outlined in the Methods Section to demonstrate the evolution of the search strategy and how it was rigorously tested. A summary of the articles from the systematic searches undertaken is given here. Across all three searches, the total number of articles was 23; this included 5 articles related to hope, 12 articles related to uncertainty, and 6 articles related to emotion regulation. A total of 1991 individuals (796 male, 1195 female, 0 unknown) with an average age of 46.1 years (n = 18/23 studies and 1866 participants) were considered. This broke down to include a total of 246 individuals (116 male, 130 female, and 0 unknown) who were included in the first search exploring the concept of hope (See Table 1). A total of 1070 individuals (379 male, 691 female, and 0 unknown) were included in the second search on uncertainty (See Table 2). A total of 675 individuals (301 male, 374 female, and 0 unknown) were included in the final search exploring the concept of emotion regulation (see Table 3). Only 8 of the 22 studies specified the ethnicity of participants. Data was obtained from a range of countries, with the USA being the most common (*n* = 7), followed by the UK (*n* = 4). Lastly, the most common qualitative methodology used was semi-structured interviews.

**Table 1 behavsci-15-01399-t001:** Social demographics for the articles from the first search on hope.

Article	Country	Gender		Age	Ethnicity of Sample	Time with Condition (CLBP)	Methodology
[18] ([18])	UK (Keele University)	Male	15	Range: 19–59 years Mean: Not stated.	Not reported	12+ weeks	Semi-structured interviews
		Female	22				
		Unknown	0				
[47] ([47])	Denmark	Male	8	Range: 28–79 years Mean: Not stated.	Not reported	Any duration of non-specific LBP—the study did not restrict inclusion based on pain duration, nor specify exact duration for each participant.	Semi-structured interviews pre- and post-consultation. Setting: Primary Care
		Female	10				
		Unknown	0				
[82] ([82])	USA	Male	8	Range: 66–83 years Mean: 56 years	Non-Hispanic Caucasian	12+ weeks	Semi-structured 1:1 interviews
		Female	13				
		Unknown	0				
[86] ([86])	UK (Oxford)	Male	7	Range: 29–67 years Mean: 52 years	Not reported	3–23 years	Semi-structured interviews (before, after, and 1-year follow-up).
		Female	13				
		Unknown	0				
[90] ([90])	Poland	Male	78	Range: Not stated. Mean: 50.45 years	Not reported	1+ year	Cross-sectional study
		Female	72				
		Unknown	0				

**Table 2 behavsci-15-01399-t002:** Social demographics for the articles from the second search on uncertainty.

Article	Country	Gender		Age	Ethnicity of Sample	Time with Condition (CLBP)	Methodology
[2] ([2])	Canada	Male	10	Range: 26–67 years Mean: 49.3 years.	Not reported	5+ years (*n* = 16) 1–5 years (*n* = 6)	Semi-structured interviews (via phone or video call).
		Female	12				
		Unknown	0				
[4] ([4])	Sweden	Male	7	Range: 15–64 years Mean: 36 years.	15 participants were born in Sweden 1 participant was born in Morocco 1 participant was born in Ethiopia	Range: 6 months–30 years. Median duration: 8 years.	Semi-structured interviews
		Female	10				
		Unknown	0				
[6] ([6])	USA	Male	9	Range: 27–70 years Mean: Not stated.	Not reported	All participants had CLBP (>3 months), but the exact duration for each participant was not specified.	Semi-structured interviews
		Female	6				
		Unknown	0				
[10] ([10])	Australia	Male	11	Range: 19–64 years Mean: 42 years	Not reported	Range: 6 months–29 years. Median duration: 7 years.	Semi-structured interviews
		Female	25				
		Unknown	0				
[19] ([19])	Australia	Male	5	Range: 21–75 years Mean: 42 years	Caucasian: 9 Latino: 2 Asian: 1 Mixed: 3	2–5 years (*n* = 5) >5 years (*n* = 10)	Semi-structured interviews
		Female	10				
		Unknown	0				
[22] ([22])	Australia	Male	16	Range: 19–85 years Mean: Not stated.	Not reported	<3 months: 4.6% 3 months to 1 year: 6.1% 13 months to 5 years: 10.8% 6–10 years: 13.9% 11–20 years: 29.2% Over 20 years: 30.8%	Ethnographic observations
		Female	49				
		Unknown	0				
[25] ([25])	USA	Male	149	Range: 19–65 years Mean: 39.8 years	White: 81.8% Black: 7.4% Asian: 0.3% Native American: 3.9% Hispanic: 6.3% Other/Unknown: = 1.8%	>3 months	Quantitative research design involving a retrospective chart review
		Female	192				
		Unknown	0				
[42] ([42])	Finland	Male	0	Range: 20–66 years Mean: Not stated	Not reported	>3 months	Qualitative: narrative analysis
		Female	30				
		Unknown	0				
[48] ([48])	USA	Male	30	All > 65 Years Mean: 83 years	Caucasian: 51% African American: 37% Hispanic: 11% Other/multiracial: 10%	5–10 years 26% >10 years 55%	Semi-structured interviews
		Female	63				
		Unknown	0				
[62] ([62])	UK	Male	3	Range: 32–53 years Mean: 45 years	White	6–18 years	Semi-structured interviews
		Female	2				
		Unknown	0				
[72] ([72])	UK	Male	129	All were >18 years. Range not stated. Mean: 49.03 years	Not reported	>3 months	Cross-sectional study
		Female	284				
		Unknown	0				
[83] ([83])	Canada	Male	10	Range: 22–63 years Mean: 47.7 years	Not reported	3–6 months	Semi-structured interviews.
		Female	8				
		Unknown	0				

**Table 3 behavsci-15-01399-t003:** Social demographics for the articles from the third search on emotion regulation.

Article	Country	Gender		Age	Ethnicity of Sample	Time with Condition (CLBP)	Methodology
[29] ([29])	USA	Male	53	Range: 18–70 years Mean: 46.3 years	Caucasian: 80% (*n* = 84) African American: 15.2% (*n* = 16) Hispanic: 4.8% (*n* = 5)	All participants had LBP for a minimum 6 months. Average duration: 9.04 years	Cross-sectional study
		Female	51				
		Unknown	0				
[38] ([38])	France	Male	120	Range: 21–61 years. Mean: 41.74 years	Not reported	<1 year (*n* = 25) 1–5 years (*n* = 107) >5 years (*n* = 124)	Semi-structured interviews
		Female	136				
		Unknown	0				
[54] ([54])	Romania	Male	17	Range: 27–84 years Mean: 50 years	Not reported	Acute LBP (*n* = 15) * Chronic LBP (*n* = 31)	Cross-sectional study
		Female	29				
		Unknown	0				
[55] ([55])	Spain	Male	15	Range: 21–64 years Mean: 49.2 years.	Not reported	12–80 weeks Mean duration: 46.5 weeks	Semi-structured interviews
		Female	39				
		Unknown	0				
[85] ([85])	USA	Male	86	Range: 18–80 years Mean: 44.05 years	Non-Hispanic Black: *n* = 115 (62.5%) Non-Hispanic White: *n* = 69 (37.5%)	3 to 6 months: 4.4% 6 months to 1 year: 6.6% 1 to 3 years: 16.9% 3 to 5 years: 18.6% 5 to 10 years: 23.5% 10 to 20 years: 13.0% Over 20 years: 7.1%	Cross-sectional study
		Female	97				
		Unknown	0				
[93] ([93])	USA	Male	10	All 18+ years. Range not detailed. Mean: 36.9 years	Caucasian American: 74.0% African American: 14.0% Asian/Asian American: 2.9% American Indian/Alaskan Native: 0.8% Native Hawaiian/Other Pacific Islander: 0.4% Other race: 7.4% Hispanic/Latino (across all races): 6.6% Non-Hispanic: 93.0%	>3 months	Cross-sectional study
		Female	22				
		Unknown	0				

* Note: Chronicity duration was not explicitly stated ([54]).

### 3.2. Quality Considerations

Quality scores for the included articles are considered in Table 4 below. This assessment, as required by this type of review, indicates that no study should be excluded and that all articles will be useful for supporting the idea-generating process.

**Table 4 behavsci-15-01399-t004:** Quality cores for originally included empirical studies.

Quality Scores for Originally Included Empirical Studies Exploring the Concept of Hope:					
Article	(a) Are Considerations and Information Given by the Selected Articles Made Sufficiently Well so That Concepts Can Be Translated?	(b) Do Findings Provide a Context for the Culture, Environment, and Setting?	(c) Are the Findings Relevant and Useful Given the Focus or Aims of the Analysis Now?	(d) Do the Questions Asked or Aims from the Paper Selected Align to Those Sought by the Meta-Ethnographer?	(e) To What Extent Do the Findings Give Theoretical Insight and Context of Interpretation Made?
[18] ([18])	Yes	Yes	Yes	Yes	To a large extent
[47] ([47])	Yes	Yes	Yes	Yes	To a large extent
[82] ([82])	Yes	Partially—limited ethnic diversity. Focus was on a specific geographical location/population.	Yes	Yes	To a large extent
[86] ([86])	Yes	Yes	Yes	Yes	To a large extent
[90] ([90])	Yes	Partially—cultural context is not deeply explored.	Yes	Yes	Moderate–large extent
Quality scores for originally included empirical studies exploring the concept of uncertainty:					
[2] ([2])	Yes	Yes	Yes	Yes	To some extent—focused on living with pain during COVID-19 pandemic.
[4] ([4])	Yes	Partially—cultural context is not deeply explored.	Yes	Yes	To a large extent
[6] ([6])	Yes	Yes	Yes	Yes	To a large extent
[10] ([10])	Yes	Yes	Yes	Yes	Moderate–large extent
[19] ([19])	Yes	Yes	Yes	Yes	To a large extent
[22] ([22])	Yes	Yes	Yes	Yes	To a large extent
[25] ([25])	Yes	Yes	Yes	Yes	Moderate extent
[42] ([42])	Yes	Yes	Yes	Yes	To a large extent
[48] ([48])	Yes	Yes	Yes	Yes	To a large extent
[62] ([62])	Yes	Yes	Yes	Yes	To a large extent
[72] ([72])	Yes	Yes	Yes	Yes	To a large extent
[83] ([83])	Yes	Yes	Yes	Yes	Moderate extent—the focus was on returning to work, but the categories of perceived uncertainty are highly relevant and in keeping with our broader findings.
Quality scores for originally included empirical studies exploring the concept of emotion regulation:					
[29] ([29])	Yes	Partially—moderate detail. Does not deeply explore broader sociocultural influences.	Yes	Yes	To some extent
[38] ([38])	Yes	Partially—adequate environmental context provided but ethnic or cultural background not discussed.	Yes	Yes	To a large extent
[54] ([54])	Yes	Partially—cultural norms and environmental context are not discussed	Yes	Yes	To a large extent
[55] ([55])	Yes	Partially—cultural references not deeply analysed.	Yes	Yes	Moderate–large extent
[85] ([85])	Yes	Yes	Yes	Yes	To a large extent
[93] ([93])	Yes	Partially—sociocultural influences not deeply explored.	Yes	Yes	To a large extent

### 3.3. Proposed Substantive Theory

The findings from articles across various fields areas were used to generate a substantive theory, presented as a model (see Figure 5). This model demonstrates how emotion regulation is closely linked to—and influenced by—tolerance of uncertainty and hope. Healthcare professionals can use this model to guide the management of people with CLBP and potentially improve their health outcomes.

#### 3.3.1. The Core Interrelated Processes of Hopelessness, Cognitive Flexibility, and Intolerance of Uncertainty

The core of this theoretical model focuses on how an individual with CLBP regulates their emotions, and how their experience of the unknown is influenced by three central and interrelated factors:(a)Intolerance of uncertainty.(b)Hopelessness.(c)Cognitive flexibility.

##### Intolerance of Uncertainty

Individuals who have chronic pain and struggle to tolerate uncertainty often express concerns about the validity of medical tests and may engage in repeated activities aimed at pinpointing a specific diagnosis ([94]). The inability to tolerate uncertainty is associated with attachment anxiety (concerns about being rejected, the need for reassurance, or the perception of being insecure) and worry ([17]). The experience of worry or depression can lead individuals to anticipate negative outcomes from unknown events, and this may be further complicated by attachment orientation and anxiety ([8]; [59]) (see Section Predisposing Factors for Maladaptive ERS). These findings suggest that a vicious cycle of negative emotions and negative expectations can easily develop in individuals with CLBP as a result of intolerance of uncertainty. Identifying intolerance of uncertainty is therefore important. Given this understanding, the experience, frequency, and persistence of negative emotions are essential considerations when setting treatment goals and helping individuals tolerate distress during pain flares ([29]).

##### Hopelessness

Prior research has established that hopelessness is a common psychological state for individuals with CLBP and is a predisposing factor for depression, anxiety, and loneliness ([82]; [92]). Hopelessness comprises three components ([39]): (1) dismal expectations—the belief that future outcomes will be negative; (2) blocked goal-directed processing—the perception that one’s ability to achieve goals is consistently thwarted; and (3) helplessness—a feeling of being unable to change one’s situation or influence outcomes. Together, these components are crucial for understanding the dynamics of hopelessness and its impact on mental health and well-being ([74]). Specific screening tools and therapies designed to foster hope are therefore important to consider ([74]).

##### Cognitive Flexibility

People with CLBP can become hypervigilant toward the experience of pain. Rather than being adaptive in their thinking and behaviour—able to cope with internal or external stressors (i.e., being cognitively flexible ([44]))—they may become rigid in their thinking or cognitively inflexible ([88]). This may manifest as a greater focus on the past or excessive worry about the future, rather than being able to accept experiences and consider more positive thoughts, beliefs, or behaviours ([24]; [82]). The inability to be cognitively flexible is significant for people with CLBP ([45]) and is identified as a critical factor in the shift from adaptive to maladaptive emotional regulation strategies ([92]). Screening and managing cognitive flexibility are important, as they create opportunities for more positive outcomes, for example, through specific therapies ([24]).

#### 3.3.2. Emotional Regulation Strategies That Influence the Core Process of Emotional Regulation

This core is directly influenced by both adaptive and maladaptive regulation strategies, including predisposing factors, the Behavioural Inhibition System (BIS), and the Behavioural Activation System (BAS). The red box on the left-hand side identifies predisposing factors for maladaptive ERSs. In contrast, the green box highlights aspects that could contribute to adaptive ERSs, though further research is needed in this area.

##### Predisposing Factors for Maladaptive ERS

Several aspects of a patient’s medical history may predispose them to maladaptive emotion regulation strategies. These factors are captured in current guidelines for the management and screening of CLBP, which include an individual’s history and psychosocial risk factors (e.g., [91]; [94]). Relevant factors include emotional disorders, childhood experiences, and insights from existing models of emotional regulation.

Emotional disorders (such as anxiety or depression) are widely associated with difficulties with emotion regulation ([13]; [26]; [63]; [70]; [69]). Notably, individuals with chronic pain are three times more likely to be diagnosed with anxiety and depression than the general population ([50]). Emotional disorders may not only predispose individuals to chronic pain but may also develop as a consequence of it. This complexity is illustrated by research identifying a significant positive association between intolerance of uncertainty, depression, and pain catastrophising in people with chronic pain ([87]). This challenges the assumption that inadequate ERS stems solely from pre-existing psychological conditions. Alexithymia, a neuropsychological trait marked by difficulty identifying and expressing emotions, is another relevant factor. Individuals with alexithymia are twice as likely to develop CLBP, as emotional awareness is crucial for adapting to stress and making everyday decisions ([26]; [38]).

Extensive literature highlights a strong association between adverse childhood experiences—including emotional, physical, sexual, and substance abuse, early parental loss, and parental psychopathology—and chronic pain. More recently, emotion dysregulation, often disrupted during childhood, has been identified as a central psychological process in the experience of pain. A secure, healthy attachment with a caregiver during childhood is fundamental to developing effective emotion regulation skills in adulthood. Insecure attachment is linked to worse pain experiences, increased disability, and higher distress ([17]; [93]). It is also a risk factor for psychopathology and maladaptive behaviours, which are positively associated with worry and intolerance of uncertainty ([17]).

##### Aspects That Could Contribute to an Adaptive ERS

Other considerations for predisposing factors include insights derived from models relating to emotions, such as the Emotion Dysregulation Model ([69]) and the Behavioural Inhibition System ([89]). The Behavioural Inhibition System is associated with suppression and avoidance strategies, hopelessness, and higher levels of depression and pain ([34]; [73]). Conversely, this system also shows a strong negative association with cognitive reappraisal—an adaptive emotion regulation strategy—which mediates the relationship between the Behavioural Inhibition System and negative affect ([73]).

Finally, interoceptive processes (how the brain interprets internal sensory signals) may influence how individuals prone to anxiety perceive interactions with healthcare professionals ([65]). This underscores the importance of considering both psychological and physiological factors in pain management.

#### 3.3.3. The Outer Rim and Named “Unknowns” Identified by People with CLBP

The outer rim of the model highlights four key themes that people with CLBP commonly report feeling uncertain about.

##### Their Beliefs About the Unknown

Uncertainty is increasingly understood as an emotion, shaped by ongoing cognitive appraisal of what is known versus unknown, leading to either positive or negative affective responses ([27]). Emotions like fear—defined as a protective response to identifiable threats—can intensify uncertainty ([70]). The broader concept of fear of the unknown includes physiological and emotional reactions to ambiguity, influenced by factors such as past experiences, perceived importance, time, and context ([13]). Trait fear is shaped by history, while state fear reflects both trait and situational influences.

[13] ([13]) linked fear of the unknown to constructs like intolerance of uncertainty, emotion regulation, attachment, and neuroticism. Evidence shows that individuals with emotional disorders often exhibit heightened fear of the unknown. It is important to note that the experience of fear, as well as intolerance of uncertainty, relates to an individual’s ability to predict and control events, which is important for how individuals manage their CLBP ([3]; [33]). When uncertainty is appraised as threatening, it activates the Behavioural Inhibition System and thus increases the perception of negative outcomes ([13]), rather than producing a balanced appraisal that reduces avoidance and fear ([18]). Indeed, fear avoidance remains a key mechanism in persistent pain and a common maladaptive emotion regulation strategy ([69]).

Addressing dysregulated beliefs about the unknown involves identifying the nature of perceived threats ([27]; [83]). These threats are linked to hope, and there is a need to appreciate that concerns relate to what is hoped for on multiple levels—from existential concerns to social identity and daily activities ([78]). These layers are evident in qualitative studies on hope-related fears in chronic LBP ([18]). Initial interventions should focus on deconstructing fears by clarifying the unknown elements or addressing specific concerns. For instance, [14] ([14]) found that addressing fears and setting realistic goals is crucial when a clear diagnosis is unavailable. Strategies to manage uncertainty include social comparisons, self-management, and psychotherapeutic approaches ([1]; [50]; [80]; [92]). However, it remains unclear whether these reduce dysregulated beliefs or simply improve tolerance ([13]).

##### The Clinical Encounter and Diagnosis

During interactions and clinical encounters, people with CLBP identified that the most important factor to be acknowledged was having their experiences and symptoms taken seriously ([42]). One reason for this is the perception that they may not be believed ([86]; [68]). As a result, people with CLBP often attempt to legitimise their symptoms by seeking further input, such as expert opinions or diagnostic investigations ([9]; [42]). Related to this is the perception among patients that they have received a thorough examination ([66]). However, not all CLBP consultations result in positive outcomes, as the clinical encounter can pose either a threat or an opportunity ([14]).

[47] ([47]) identified different experiences before and after a consultation for people with CLBP. Before, individuals reported concerns, a sense of helplessness, and fear. Following the consultation, reduced uncertainty, a greater sense of hope, and improved emotional regulation were observed. Trust appeared to arise through examination thoroughness, and the clinician’s emotional and personal engagement also played a crucial role in the identified outcomes. The concept of trust is supported by interactions being perceived as open and honest ([68]). Trust is especially important if individuals have attachment insecurity ([23]), and this may require clinicians to be able to manage worries and concerns about the future ([85]).

The clinical encounter also needs to consider how and what advice is given. For example, when people with CLBP are advised to rest or avoid aggravating activities, it can trigger fear-avoidant behaviour. In contrast, those who report receiving adequate reassurance, support, and explanations from their clinician feel empowered and are more prepared to make appropriate lifestyle adaptations to self-manage their condition ([85]).

People with CLBP strive for a diagnosis or, at the very least, an explanation for their symptoms ([4]; [6]; [10]; [62]). Diagnostic uncertainty causes increased emotional distress, disability, and pain ([72]). It can be informed by limitations in biomedical knowledge (medical uncertainty) and a more general awareness among patients of an undetermined future (existential uncertainty) ([21]). Several qualitative studies have found that, typically across musculoskeletal care, clinical practice focuses on ruling out serious pathology rather than ruling in a definitive diagnosis ([14]).

However, within society, there is a strong perception and expectation that a thorough examination—sometimes involving diagnostic investigations such as blood tests or imaging—will lead to a legitimate diagnosis, which in turn enables effective treatment interventions and hope for the resolution of symptoms ([10]; [22]). Despite this understanding of patient expectations, national guidelines do not recommend imaging for low back pain with or without radicular symptoms in the absence of red flags or a neurological deficit ([57]). This is because, in most cases (90–95%), there is no identifiable structural cause in people with CLBP, meaning that pathological findings from imaging frequently do not correlate with the individual’s symptoms ([22]).

It is well established that in cases where there is no clear diagnosis or explanation for symptoms, this can negatively impact pain, disability, and both cognitive and emotional functioning ([9]; [72]). Additionally, when a structural cause is not identified through imaging, it can result in a contradictory outcome because it still fails to provide a clear diagnosis. At times, incidental findings may occur, which foster further anxiety and fear, or even cause the individual to question their pain experience—ultimately failing to improve outcomes ([19]).

Clinicians who provide clear, consistent, and empathetic explanations during CLBP consultations help to reduce uncertainty, improve patient satisfaction, and enhance prognostic outlook ([47]; [52]).

Despite this, astonishingly, there remains no clinical guidance on how to manage patients in the face of uncertainty ([52]). Uncertainty is becoming an increasingly prevalent issue from the patient’s perspective due to heightened public awareness of the limitations of medical knowledge—disseminated through various media channels—which contributes to increased anxiety around health and illness risk in society ([20]; [41]).

Physiotherapists, in particular, struggle to manage diagnostic uncertainty when treating people with CLBP because they feel they lack sufficient knowledge and skills, as well as time and resources, to manage the condition’s complexity effectively ([21]). As a compensatory strategy, healthcare professionals may avoid or minimise discussions around uncertainty to maintain the perception of expertise or avoid compromising their authority ([21]). However, a lack of acknowledgement or openness can undermine trust and may result in epistemic injustice ([22]). Epistemic injustice occurs when healthcare professionals use their authority to influence a patient’s decision to align with their own, or when a patient’s personal account is discredited ([68]).

Different professional groups manage uncertainty in diverse ways ([41]; [46]). [53] ([53]) outlined two contrasting approaches to managing uncertainty: the logic of care and the logic of choice. In Western clinical contexts, healthcare professionals often prioritise offering the individual a choice by outlining factual information. However, this approach assumes that the individual possesses high self-efficacy—that is, they feel competent, confident, and capable of contributing to decisions and managing their own condition.

In contrast, people with CLBP often exhibit low self-efficacy and may not always respond positively to choice, particularly when faced with navigating multiple uncertain options. In such cases, [53] ([53]) advocates the logic of care, which promotes a patient-centred approach by focusing on relevant, achievable goals and supporting individuals to make lifestyle adaptations that enable them to self-manage their condition effectively ([3]).

##### The Impact on Self-Identity, Social Relationships, and the Future

The expression of hope is associated with self-identity, social relationships, and future aspirations, all of which are identified as meaningful ([74]). A common theme across several qualitative studies is that people with CLBP frequently experience self-doubt in their ability to manage pain and navigate daily challenges ([10]; [18]). This impacts their social identity, relationships, and ability to envision a meaningful future. Their (in)ability to manage fluctuating symptoms threatens their sense of self, creating fear about the future ([62]).

The beliefs of an individual’s social network such as family, friends, or colleagues can also influence expectations, attitudes, and overall prognosis, although to a lesser extent than the opinions and beliefs of clinicians ([10]). Chronic pain can have profound social consequences ([62]). It is often associated with increased social isolation, which can lead to depression and feelings of lost purpose and value ([2]). Social isolation has been identified as a more significant consequence for older adults living with CLBP compared to younger adults, but it is not commonly assessed in clinical practice ([48]). Qualitative literature ([62]) has revealed that people with CLBP tend to withdraw from social contact for several reasons: to avoid symptom flares, they perceive themselves as unacceptable company, they prefer the safety of solitude, and because of the stigma and shame associated with chronic pain. Furthermore, diagnostic uncertainty has been found to correlate positively with guilt. When people cannot find a cause for their pain, they may blame themselves and experience social guilt—fear of letting others down—which is strongly associated with anxiety and depression ([72]).

[86] ([86]) found that maintaining a positive self-identity was a key factor in the success of individuals enrolled in a pain management programme for CLBP. It also plays an important role in helping them overcome fears related to movements or activities that might trigger or worsen their symptoms ([86]). Therefore, it is essential that clinicians explore what is personally meaningful to each individual in order to set relevant and motivating goals ([78]).

Clinicians should use active listening and risk stratification tools to determine where individuals with CLBP fall on the spectrum from entrenched self-doubt to effective self-management, and then tailor an appropriate action plan ([57]). It is also important that healthcare professionals understand how pain-related behaviour patterns—such as fear avoidance or avoidance–endurance—can influence cognitive rigidity and require different treatment approaches ([53]). Fear-avoidant behaviours are addressed through Interdisciplinary Multimodal Pain Therapy (IMPT), which includes correcting misconceptions and using exposure-based interventions. In contrast, avoidance–endurance behaviours require a different theoretical focus: encouraging individuals to recognise warning signs of mental and physical deterioration and to adopt pacing strategies ([53]).

##### Treatment Failure

Failed treatments and the inability to self-manage pain can be another source of uncertainty, particularly when individuals have not achieved treatment goals despite adherence ([10]). This often triggers negative emotions and is reported to lead people to feel “powerless” and “helpless” ([82]).

People with CLBP often receive extensive input from healthcare services, but treatment efficacy is variable, frequently leading to frustration with healthcare systems ([25]; [52]). There is increasing recognition that the success of adopting a self-management approach hinges on readiness for change and acceptance of pain ([25]). Acceptance appears to be a key adaptive emotion regulation strategy, enabling individuals to find ways to live a fulfilling life despite pain ([82]). Thus, recommendations for therapies that support acceptance could be considered.

#### 3.3.4. The Model Output and Resultant ERSs

The model’s output focuses on ERSs, which are divided into two types. The first is adaptive ERS, whereby individuals effectively regulate their emotions using strategies such as acceptance, reappraisal, and problem-solving, although there is generally less research exploring this approach. In contrast, more extensive research has examined maladaptive ERSs, which involve behavioural strategies such as suppression, rumination, avoidance or reassurance-seeking, which are typically ineffective and often contribute to the development of psychopathology ([69]).

## 4. Discussion

To the best of the authors’ knowledge, this is the first example of a social constructivist meta-ethnography study conducted with the aim of generating a substantive theory. The resulting multifaceted model places emotion regulation as a central concept in people with CLBP, influenced by three interrelated concepts. The findings demonstrate that maladaptive ERSs are driven by heightened intolerance of uncertainty and hopelessness, which contribute to and interact with cognitive rigidity. Together, these concepts likely cause significant psychological distress and poor health outcomes ([30]; [44]). Importantly, improving cognitive flexibility appears to be a potential resilience factor, enabling individuals to reframe distressing experiences or uncertainty and shift from maladaptive to adaptive ERSs ([91]).

These central concepts are further influenced by intrinsic factors such as a history of emotional disorders (especially anxiety and depression), personality disorders, or personality traits that foster worry and a tendency to catastrophise pain. They are also shaped by a history of adverse childhood experiences or later-in-life adult attachment insecurity. Extrinsically, the clinical encounter plays an equally pivotal role in identifying contributing factors and creating an individualised management plan to improve emotion regulation and ultimately support self-management of CLBP.

All CLBP consultations involve an element of uncertainty for both the patient and clinician, but these encounters play a vital role in either reinforcing or alleviating that uncertainty. Establishing an effective therapeutic relationship built on trust, promoting epistemic humility, and showing compassion and a willingness to help will shape patients’ beliefs, self-efficacy, and ability to engage in a biopsychosocial management plan. Recognising the limits of healthcare and science and communicating this information effectively should be fundamental skills for all healthcare professionals ([41]).

Clinical examinations and interventions often emphasise technical factors, leaving little space for the social and political dimensions that also matter ([71]). Therefore, [20] ([20], [21]) call for more research and training on medical uncertainty, urging the development of training interventions that change our learning approach and foster an epistemic cultural shift towards a system that integrates human uncertainties into evidence-based practice. Importantly, these principles and changes must extend beyond individual clinical practice and permeate policies, procedures, and the values of our healthcare systems. After all, providing too much certainty or neglecting the issue can be unsafe, but attending to uncertainty is always optimal because “risks always involve uncertainty, but uncertainty does not always involve risks” ([21]).

### 4.1. Clinical Recommendations

This model is intended to be used as a quick reference tool for clinicians navigating the central concepts of CLBP care. Consultations offer vital opportunities to uncover the key factors driving maladaptive ERS and to offer a personalised approach. During each consultation, clinicians should focus on the following:Building an effective therapeutic relationship, founded on trust, openness, and honesty.Conducting a thorough exploration—not only of the patient’s medical history but also sensitively exploring whether they may have been affected by adverse childhood experiences and determining their attachment orientation.Exploring the patient’s beliefs about the unknown, including their perceptions of pain and worries or concerns about living with a chronic condition, and gaining a deeper understanding of their self-identities while remaining open to other possible psychosocial factors that are not captured in this study.
Clinicians should also be aware of clinical assessment tools and potential therapies that provide a useful starting point for understanding the patient.

### 4.2. Clinical Implications for Screening

Based on current research findings, we recommend specific screening for predisposing factors, supported by current guidelines (e.g., [59]; [91]). The following brief, validated scales would also be suitable for use in time-limited clinical settings:Cognitive Flexibility Scale ([49]): A 12-item measure using a 6-point Likert scale to assess an individual’s ability to adapt thinking and consider alternative solutions.Model of Emotions, Adaptation and Hope (MEAH) ([74]): A 5-item scale designed to identify an individual’s most significant named challenge. It can be administered in approximately 30 s. The hope item is particularly useful for identifying experiences of uncertainty, possibility, and hopelessness.Intolerance of Uncertainty Scale—Short Form ([5]): A 5-item measure that efficiently screens for intolerance of uncertainty.Emotion Regulation Questionnaire ([31]): A 10-item scale assessing two key strategies—cognitive reappraisal and expressive suppression—for clinicians seeking a deeper understanding of emotion regulation.

### 4.3. Clinical Implications for Therapy

When screening identifies challenges in cognitive flexibility, emotion regulation, or hope, several therapeutic approaches may be beneficial. For rehabilitation therapists, brief tools have shown to be effective:MEAH-based therapeutic conversations ([74]): These can be delivered in 10 or 30 min formats by trained rehabilitation therapists (training is available online in under an hour). The MEAH tool serves as a foundation for exploring emotional adaptation and fostering hope. The extended version may be particularly useful, as it considers social identity, relationships, and meaningful hopes as part of a structured conversation.

Other therapies with positive results should also be considered. Three examples include the following:Acceptance and Commitment Therapy (ACT) ([24]): This type of therapy focuses on improving cognitive flexibility and helping individuals accept and embrace feelings and thoughts while committing to action. It is effective at improving social and physical functioning, enhancing mood, and lowering pain.Emotional Awareness and Expression Therapy ([43]): This type of therapy helps individuals process and express avoided emotions, particularly those linked to trauma or chronic pain.Dialectical Behaviour Therapy (DBT) ([60]): This type of therapy offers structured skills training in emotion regulation, distress tolerance, and mindfulness, and has been shown to be effective in chronic pain populations.

### 4.4. Clinical Case Example

We provide a worked case example below (see Table 5) illustrating how the model could be applied in clinical practice.

This case illustrates how the model can help to (a) guide screening and assessment of psychosocial factors; (b) inform tailored interventions based on individual profiles; (c) support therapeutic conversations that foster hope and emotional adaptation; and (d) enhance clinical decision-making in complex CLBP cases.

### 4.5. Future Research

Firstly, the utilisation of this model needs to be tested within clinical practice before it can be used as a navigation tool for clinicians to improve their skills and strategies in managing patients with CLBP via a tailored approach. Other clinical areas to focus on in future research include the need for more investigation into interventions that address cognitive flexibility and cognitive reappraisal via the Behavioural Inhibition System or Behavioural Activation System, in addition to exploring ways to enhance secure attachment traits. Additionally, there is a need for the development of clinical guidelines on managing uncertainty, promoting epistemic humility from undergraduate education through to evolving healthcare policies that embrace uncertainty.

### 4.6. Limitations

Despite searching for studies involving participants with low back pain of any duration, most of the existing literature has explored these key concepts using participants with CLBP (>3 months duration). Therefore, we cannot confidently say that this model and its findings are applicable to people with acute LBP, although further testing in this group and other related populations would be warranted.

Additionally, the low number of studies may have restricted the process of idea generation, and this study may not have captured all the literature on hope, uncertainty, and emotion regulation, nor included every factor that could influence these feelings. For instance, this could include aspects such as interoceptive active inference or impaired reward-related learning signals.

Importantly, social constructivist meta-ethnography was designed to create substantive theory, not generalisable theory. Thus, the resultant model requires critical consideration and further research. As reflected by the quality scores (see Table 4 in, Section 3.2 Quality Considerations), the initial studies and, subsequently, the model may not account for multiple cultures or sociodemographic factors, which arguably could be the case for any theory. However, this is particularly important for this review due to the limited ability to apply critical enquiry from social constructivist grounded theory and reveal specific cultural or ethnic influences within the model. Therefore, further research is required to test this theory within clinical practice and across different cultures, contexts, and settings.

We acknowledge that we have focused on specific frameworks of key concepts and that our understanding may be limited, as well as the possibility that we have missed key literature supporting this understanding. Additionally, we are aware of the potential for confirmation bias, given the past research of the supervising author—seeking and confirming existing thoughts may have influenced the proposed model.

Further consideration of counselling-based literature and therapies such as Acceptance and Commitment Therapy could be relevant to the model and its application. Other psychological constructs may also play an important role, such as self-efficacy theory and sources of self-efficacy. However, it is beyond the scope of this review to address these considerations, which require further investigation.

Lastly, the application of this theory beyond people with CLBP may be limited and will require additional research.

### 4.7. Conclusions

To conclude, tolerance of uncertainty and hope are complex phenomena that significantly impact emotion regulation and health outcomes in people living with CLBP. They also have broader implications for society, warranting extensive research. The rising prevalence of CLBP demonstrates that our current approach to consultations and treatment interventions is insufficient, and a new approach to managing this population is urgently needed.

## Figures and Tables

**Figure 5 behavsci-15-01399-f005:**
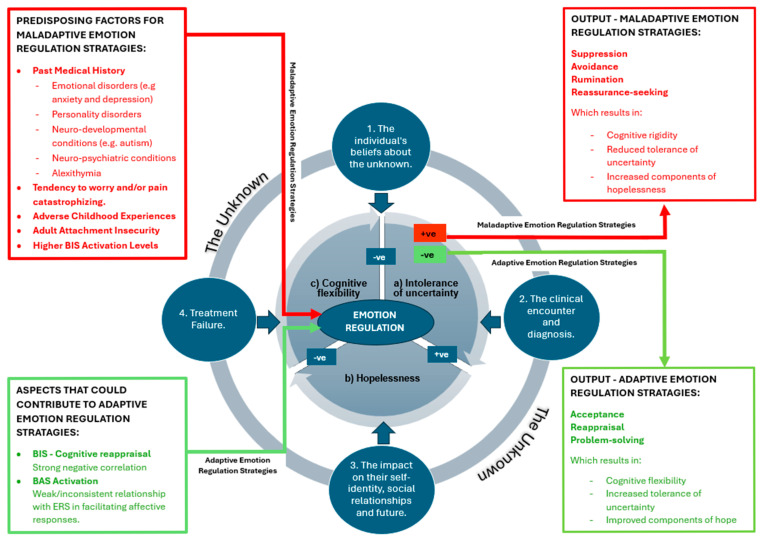
The final proposed model demonstrating how hope, tolerance of uncertainty, and emotion regulation are interconnected and influenced in individuals with CLBP.

**Table 5 behavsci-15-01399-t005:** Providing an illustrative case example.

Consideration	Example
**Patient Profile**	**Name:** Sarah (pseudonym). **Age:** 52. **Occupation:** Primary school teacher (currently on long-term sick leave). **Condition:** Chronic low back pain (duration: 6 years). **History:** Diagnosed with fibromyalgia 3 years ago; history of childhood emotional neglect; previous diagnosis of anxiety and mild depression.
**Presenting Issues**	Sarah reports: Increased pain intensity over the past 6 months. “I have been managing for a long time now, but in the last six months my social life has suffered, my pain has got worse and I need something to change. She identifies feelings of hopelessness: “At this point in time, if feels all too much, I don’t see how this will ever get better.” There are clear signs of avoidance of movement due to fear of exacerbating pain. “I have attempted some movement for exercise, and it seems to make it worse” and frustration with repeated inconclusive diagnostic tests. “I have seen five health care professionals, some from my own money, and no one can tell me what is wrong”. These problems are worsened by her difficulty trusting healthcare professionals due to past dismissive encounters. “When you are dismissed because they [health care professionals] cannot understand what is happening it’s hard to trust people in charge of your care”. She also reports social withdrawal and strained relationships with family. “I avoid family situations as I don’t want to go over the same conversations and not be believed”
**Clinical Assessment Using the Model**	**1. Screening Tools Used** MEAH Scale: Identified the greatest challenge as not being able to live a life that allowed her some form of life. Scoring low on the hope, energy and feeling scale as well as not being able to accept the current situation. Intolerance of Uncertainty Scale (IUS-5): high score indicating significant discomfort with ambiguity. Emotion Regulation Questionnaire: high expressive suppression, low cognitive reappraisal. Cognitive Flexibility Scale: low score, indicating rigid thinking patterns. **2. Model Mapping** Core Processes: Intolerance of Uncertainty: Sarah expresses distress over not having a clear diagnosis and fears the future. Hopelessness: she feels blocked in achieving goals and helpless when it comes to managing her condition. Cognitive Inflexibility: she struggles to adapt her thinking, often ruminating on past failed treatments. Maladaptive ERS: Suppression, avoidance, reassurance-seeking. Fear-avoidance behaviour reinforced by prior clinical advice to “rest.” Predisposing Factors: Childhood emotional neglect. History of anxiety and depression. Insecure attachment traits. Clinical Encounter: Prior experiences of epistemic injustice. Distrust in medical professionals. Desire for validation and thorough examination.
**Therapeutic** **Approach**	**1. Initial Goals** Rebuild trust and therapeutic alliance. Address maladaptive ERS and intolerance of uncertainty. Foster hope and cognitive flexibility. **2. Interventions** MEAH-based therapeutic conversation (30 min format): explored Sarah’s meaningful hopes and identity. Acceptance and Commitment Therapy (ACT): focused on values-based goal setting and cognitive diffusion. Pacing strategies: introduced to counter avoidance–endurance behaviour. Education on uncertainty: Normalised the experience and introduced the concept of epistemic humility.
**Outcomes After** **6 Sessions**	Sarah reported an increased sense of control and reduced fear of movement, “I have been able to get out more, I have seen my family a couple of times now”. She identified a re-engagement with light physical activity “I am managing to move without pain at times, which I didn’t think was possible before”. She identified improved emotional expression and reduced rumination. Finally, she stated a renewal of hope: “I’m starting to believe I can live well with this.”

## Data Availability

The original contributions presented in this study are included in the article/Appendix A.

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
