# Peer review of "Exploration of How Uncertainty Tolerance, Emotion Regulation, and Hope Are Linked and Influenced in People with Chronic Low Back Pain: A Worked Example of a Social Constructivist Meta-Ethnography Study"

_behavsci, 2025, doi:10.3390/bs15101399_

Round 1

Reviewer 1 Report (Previous Reviewer 2)

Comments and Suggestions for Authors

Abstract

The abstract should offer a complete interpretation of the study's results. Phrases such as "see Figure 4" should be avoided, as the abstract should stand alone without requiring reference to figures or tables.

Introduction

The authors should spell out all abbreviations in full upon first use—for instance, “HCPs” should be expanded to “healthcare professionals” when it first appears.

In its current form, the introduction is overly lengthy and includes several sections that do not directly relate to the main focus of the study. The authors are encouraged to streamline this section to clearly present the research problem, its significance, and the study’s objectives in a more concise and focused manner.

Methods and Materials

The supplementary materials was not provided.

On page 8, line 260, there is a typographical error that should be corrected.

Throughout the manuscript, several abbreviations are introduced without being spelled out on first use, which can make the text difficult to follow. The authors should ensure that all abbreviations are defined upon first mention.

Results and Discussion

In Table 1, the description of the study population would be more informative if it included additional variables such as social, psychological, and environmental factors, along with medication and surgical history. These aspects likely contribute to patients’ experiences of uncertainty, hope, and emotion regulation, and should be reported accordingly.

Regarding Figure 4, the inclusion of “cognitive flexibility” as a core construct in the theoretical model is problematic. This concept is not introduced or discussed in the main text, raising questions about its relevance and justification. Moreover, several terms used in the figure do not appear elsewhere in the manuscript, making the model difficult to interpret. The theoretical framework lacks a clear logical progression and does not appear to be directly supported by the reviewed literature. For example, medical history is incorporated into the model, yet it is not discussed or summarized in the article review table. This raises concerns about the evidence base for the model.

Additionally, when discussing potential predisposing factors that influence emotion regulation, many of the studies cited are not part of the included review articles. If these factors are considered central to the model, they should be supported by a more comprehensive literature review.

Also, “cognitive flexibility” was not included as a keyword in the literature search strategy. Without a thorough review of studies related to this construct, its inclusion in the theoretical model appears unsubstantiated.

The authors did not provide a clear summary or discussion of how factors such as uncertainty, hope, and emotion regulation influence treatment outcomes, based on the findings of the included review articles.

Comments on the Quality of English Language

The manuscript contains several typographical errors that should be carefully reviewed and corrected

Author Response

Reviewer 1

Author response: Thank you for your comments and the time taken to consider the review article.

Abstract

The abstract should offer a complete interpretation of the study's results. Phrases such as "see Figure 4" should be avoided, as the abstract should stand alone without requiring reference to figures or tables.

Author response: This has been changed.

Introduction

The authors should spell out all abbreviations in full upon first use—for instance, “HCPs” should be expanded to “healthcare professionals” when it first appears.

Author response: We have reduced the number of abbreviations to clear ones.

In its current form, the introduction is overly lengthy and includes several sections that do not directly relate to the main focus of the study. The authors are encouraged to streamline this section to clearly present the research problem, its significance, and the study’s objectives in a more concise and focused manner.

Author response: We have streamlined the section in order to make it more concise and focused on the main area of study. 

Methods and Materials

The supplementary materials was not provided.

Author response: Apologies we believe it was previously submitted. We have ensured it is attached this time.

On page 8, line 260, there is a typographical error that should be corrected.

Author response: This has been checked and rewritten. Thank you.

Throughout the manuscript, several abbreviations are introduced without being spelled out on first use, which can make the text difficult to follow. The authors should ensure that all abbreviations are defined upon first mention.

Author response: Thank you we agree and have decided to keep just two abbreviations to address this and ensure the reader doesn’t have to keep checking what abbreviations are referring to.

Results and Discussion

In Table 1, the description of the study population would be more informative if it included additional variables such as social, psychological, and environmental factors, along with medication and surgical history. These aspects likely contribute to patients’ experiences of uncertainty, hope, and emotion regulation, and should be reported accordingly.

Author response: Although we agree adding in these variables would be valuable, we have checked all included articles for medication and surgical history and none of the studies outline this information in participant characteristics tables. In some studies, these factors are discussed in narrative format due to the qualitative nature of the studies.

This type of clinical information would be commonly found in quantitative studies examining treatment outcomes. It is therefore acknowledged as a limitation of the model.

Regarding Figure 4, the inclusion of “cognitive flexibility” as a core construct in the theoretical model is problematic. This concept is not introduced or discussed in the main text, raising questions about its relevance and justification. Moreover, several terms used in the figure do not appear elsewhere in the manuscript, making the model difficult to interpret. The theoretical framework lacks a clear logical progression and does not appear to be directly supported by the reviewed literature. For example, medical history is incorporated into the model, yet it is not discussed or summarized in the article review table. This raises concerns about the evidence base for the model.

Author response: We have introduced it in the main text as it is a key factor that affects hope. We agree we need to demonstrate how we identified the concept and what searching was undertaken around them.

Additionally, when discussing potential predisposing factors that influence emotion regulation, many of the studies cited are not part of the included review articles. If these factors are considered central to the model, they should be supported by a more comprehensive literature review.

Author response: we have identified how these factors were identified and searched.

Also, “cognitive flexibility” was not included as a keyword in the literature search strategy. Without a thorough review of studies related to this construct, its inclusion in the theoretical model appears unsubstantiated.

Author response: We have identified searches undertaken on concepts, we have identified (1) how the concepts were identified for the model (2) how the searches were undertaken for each concept.

The authors did not provide a clear summary or discussion of how factors such as uncertainty, hope, and emotion regulation influence treatment outcomes, based on the findings of the included review articles.

Authors response: we agree we have identified this now.

Reviewer 2 Report (Previous Reviewer 3)

Comments and Suggestions for Authors

I suggest that the presentation of the information be improved, specifically the scientific writing. It should be direct, simple, and concise. Throughout the manuscript, the ideas are overly elaborate, particularly in the methods and results sections, which affects clarity and comprehension.

Regarding the study design, it should be clearly defined. The authors state that they used a meta-ethnographic analysis; however, they also apply elements of a systematic review. Upon reviewing the steps of the systematic review, I noticed that they are not entirely correct—for example, the search strategy, as well as the definition of the inclusion and exclusion criteria for the articles.

Therefore, I recommend that the authors conduct the systematic review again, strictly following the steps outlined in the PRISMA guidelines.

As for the ethnographic method, it must be clearly defined, especially considering that it typically leads to the generation of theories. Concerning the proposed theory, the authors should establish a logical and coherent connection that clearly shows how they arrived at their theoretical model.

I believe that if the authors carry out a proper systematic review and then develop their ideas in a clear, simple, and direct manner, they will be better able to support their proposed theoretical model.

Author Response

Reviewer 2

I suggest that the presentation of the information be improved, specifically the scientific writing. It should be direct, simple, and concise. Throughout the manuscript, the ideas are overly elaborate, particularly in the methods and results sections, which affects clarity and comprehension.

Author response: we have attempted to make aspects more direct, simple and concise as identified.

Regarding the study design, it should be clearly defined. The authors state that they used a meta-ethnographic analysis; however, they also apply elements of a systematic review. Upon reviewing the steps of the systematic review, I noticed that they are not entirely correct—for example, the search strategy, as well as the definition of the inclusion and exclusion criteria for the articles. Therefore, I recommend that the authors conduct the systematic review again, strictly following the steps outlined in the PRISMA guidelines.

Author response: We have attempted to make changes to improve clarity and this is in direct response to other reviewers comments. We believe this has improved the manuscript. However, the framework we used supports aspects of PRISMA 2020 i.e., searching but not other aspects like quality appraisal, certainty assessment etc, so it wouldn’t be appropriate to undertake a systematic review according to PRISMA 2020 guidelines as this is for intervention studies.

As for the ethnographic method, it must be clearly defined, especially considering that it typically leads to the generation of theories. Concerning the proposed theory, the authors should establish a logical and coherent connection that clearly shows how they arrived at their theoretical model.

Author response: We are not sure why the supplementary file was not with the manuscript. We believe that this elaborates on the proposed substantive theory.

I believe that if the authors carry out a proper systematic review and then develop their ideas in a clear, simple, and direct manner, they will be better able to support their proposed theoretical model.

Author response: thanks for this comment. We have followed a framework for meta-ethnographic reviews which includes a systematic search. It wouldn’t be appropriate to undertake a systematic review as this is not needed.

Reviewer 3 Report (New Reviewer)

Comments and Suggestions for Authors

This manuscript addresses a highly relevant and increasingly studied area within chronic pain management: the interplay between uncertainty tolerance, hope, and emotion regulation in individuals with chronic low back pain (CLBP). The authors present a worked example of a social constructivist meta-ethnography, aiming to generate a substantive theory that links these psychosocial constructs. Overall, the topic is timely and potentially impactful for both theory development and clinical practice. However, there are several important aspects that require clarification or further elaboration before this work can be considered for publication.

First, while the philosophical and methodological stance is clearly stated (social constructivism; modified meta-ethnography with elements of constructivist grounded theory), the justification for using this hybrid approach could be strengthened. The choice of methodology seems appropriate given the interpretative aims, but the distinction between meta-ethnography and grounded theory elements is at times blurred. The authors should clearly articulate what was borrowed from each tradition and why.

Second, the iterative nature of the analytic process is well described, but at times it becomes difficult to follow the evolution of the inclusion criteria, particularly how the concepts of uncertainty and emotion regulation were progressively incorporated. The rationale for expanding the eligibility criteria mid-process must be made more transparent, as it raises questions about selection bias and the stability of the emerging theory. For example, including studies on acute low back pain in a review focused on CLBP, even when justified by iterative refinement, may dilute the conceptual coherence of the findings.

Third, the use of PRISMA diagrams and a registered protocol (PROSPERO) adds credibility to the process. However, given the qualitative and evolving nature of the synthesis, the audit trail and supplementary materials become essential. These should be more explicitly integrated into the main manuscript — for instance, clarifying at which steps additional literature searches were conducted, and how these shaped the emerging model.

Fourth, the model presented (Figure 4) is interesting and potentially valuable, especially in its identification of core interacting factors such as intolerance of uncertainty, hopelessness, and cognitive flexibility. However, its practical utility for clinicians remains vague. The authors refer to the potential clinical implications, but do not sufficiently articulate how this model could be translated into assessment tools, communication strategies, or interventions. Including a worked clinical application or hypothetical case scenario might enhance its translational impact.

Fifth, while the discussion of related constructs (e.g., attachment, pain catastrophizing, behavioural inhibition system) is rich, the review sometimes risks becoming overly theoretical. A tighter synthesis focusing on the most salient mechanisms that directly inform emotion regulation and hope in CLBP would improve clarity and reduce cognitive overload for the reader. Similarly, the repetition of theoretical content already well-known in the pain field could be trimmed in favour of a more original analytical narrative.

Lastly, the authors briefly touch upon the limitations of their review, but this section would benefit from more critical reflection. Issues such as potential confirmation bias, limitations in the generalisability of the model (despite claims of analytical generalisability), and the limited ethnic diversity in many included studies should be acknowledged more explicitly.

Minor comments:
The authors should ensure that the manuscript complies with the journal’s formatting guidelines. For instance, figure captions are inconsistently placed (e.g., Figure 1 has the title above, while Figure 2 has it below), and this should be standardised throughout. Additionally, Figures 2 and 3 are of relatively low visual quality and should be improved to ensure legibility and clarity, particularly if they are to be reproduced in print.

Author Response

Reviewer 3

This manuscript addresses a highly relevant and increasingly studied area within chronic pain management: the interplay between uncertainty tolerance, hope, and emotion regulation in individuals with chronic low back pain (CLBP). The authors present a worked example of a social constructivist meta-ethnography, aiming to generate a substantive theory that links these psychosocial constructs. Overall, the topic is timely and potentially impactful for both theory development and clinical practice. However, there are several important aspects that require clarification or further elaboration before this work can be considered for publication.

Author response: thank you for the comments and suggestions below.  

First, while the philosophical and methodological stance is clearly stated (social constructivism; modified meta-ethnography with elements of constructivist grounded theory), the justification for using this hybrid approach could be strengthened. The choice of methodology seems appropriate given the interpretative aims, but the distinction between meta-ethnography and grounded theory elements is at times blurred. The authors should clearly articulate what was borrowed from each tradition and why.

Author response: Given the requests from other reviews to keep the manuscript tight. We have more clearly identified and refer readers to the framework which identifies how this type of grounded theory was used to enhance the theory generating process of meta-ethnography.

Second, the iterative nature of the analytic process is well described, but at times it becomes difficult to follow the evolution of the inclusion criteria, particularly how the concepts of uncertainty and emotion regulation were progressively incorporated. The rationale for expanding the eligibility criteria mid-process must be made more transparent, as it raises questions about selection bias and the stability of the emerging theory. For example, including studies on acute low back pain in a review focused on CLBP, even when justified by iterative refinement, may dilute the conceptual coherence of the findings.

Author response: Figure 2 is expanded fully in Supplementary file and should achieve this. We have also expanded the results.

Third, the use of PRISMA diagrams and a registered protocol (PROSPERO) adds credibility to the process. However, given the qualitative and evolving nature of the synthesis, the audit trail and supplementary materials become essential. These should be more explicitly integrated into the main manuscript — for instance, clarifying at which steps additional literature searches were conducted, and how these shaped the emerging model.

Author response: Apologies we believe it was previously submitted. We have ensured it is attached this time.

Fourth, the model presented (Figure 4) is interesting and potentially valuable, especially in its identification of core interacting factors such as intolerance of uncertainty, hopelessness, and cognitive flexibility. However, its practical utility for clinicians remains vague. The authors refer to the potential clinical implications, but do not sufficiently articulate how this model could be translated into assessment tools, communication strategies, or interventions. Including a worked clinical application or hypothetical case scenario might enhance its translational impact.

Author response: This has been addressed, see section 4.1, 4.2 & 4.3

Fifth, while the discussion of related constructs (e.g., attachment, pain catastrophizing, behavioural inhibition system) is rich, the review sometimes risks becoming overly theoretical. A tighter synthesis focusing on the most salient mechanisms that directly inform emotion regulation and hope in CLBP would improve clarity and reduce cognitive overload for the reader. Similarly, the repetition of theoretical content already well-known in the pain field could be trimmed in favour of a more original analytical narrative.

Author response: Thank-you. This has been amended.

Lastly, the authors briefly touch upon the limitations of their review, but this section would benefit from more critical reflection. Issues such as potential confirmation bias, limitations in the generalisability of the model (despite claims of analytical generalisability), and the limited ethnic diversity in many included studies should be acknowledged more explicitly.

Author response: Thank you - we have expanded on this in section 4.5.

Minor comments:
The authors should ensure that the manuscript complies with the journal’s formatting guidelines. For instance, figure captions are inconsistently placed (e.g., Figure 1 has the title above, while Figure 2 has it below), and this should be standardised throughout. Additionally, Figures 2 and 3 are of relatively low visual quality and should be improved to ensure legibility and clarity, particularly if they are to be reproduced in print.

Author response: Thank-you. This has been amended.

Round 2

Reviewer 3 Report (New Reviewer)

Comments and Suggestions for Authors

Thank you for the revisions. The manuscript has improved substantially in clarity, methodological transparency, and formatting. Most concerns have been adequately addressed.

One point remains: the clinical applicability of the model could be further strengthened by adding a more concrete example (e.g., a worked case scenario) to illustrate its use in practice.

Overall, the manuscript is now much clearer and well-prepared.

Author Response

Many thanks for your feedback and suggestion of a worked case scenario to illustrate its use in practice.

Please see section 4.4 Clinical Case Example - page 30. 

This manuscript is a resubmission of an earlier submission. The following is a list of the peer review reports and author responses from that submission.

Round 1

Reviewer 1 Report

Comments and Suggestions for Authors

The authors present an interesting new approach to analyzing construct relationships that play a role in the context of chronic back pain and should be taken into account in therapy. It seems plausible to me that hope, uncertainty/ambiguity tolerance and emotion regulation play an important role in the modulation of the experience of pain and impairment and are interlinked. Understanding the method of theory development was rather difficult for me as a quantitative researcher with a background in affective/ clinical neuroscience. In my view, the value of the final model lies in the fact that it attaches great importance to emotion regulation. When the concept of emotion regulation is introduced, maybe it would be informative to differentiate between explicit and implicit emotion regulation. On Page 12 you elaborate on the role of interoceptive processes largely influencing implicit emotion regulation involving automatized/ conditioned regulation processes operating outside of conscious awareness. 

Overall, some of the connections between the constructs introduced remain unclear to me. I was wondering, for example whether “hope” is really an emotion or related to the cognitive construct of self-efficacy or action orientation (Kuhl). Can “hope” really be considered an emotion as is seems to entail several cognitive-evaluative processes?

In some sections, the line of argument could be more stringent. New complex constructs are constantly being introduced, making it difficult to derive central ideas which may inform future research.  

I find it astonishing that the connection between tolerance of ambiguity/ tolerance of uncertainty and internalized attachment experiences has been little addressed in the literature. I think it is plausible that basic trust/secure attachment plays an important role with regard to the central concepts of the model.

Overall, I think that the topic of this article and some of the ideas are very relevant to advancing pain management by addressing uncertainty intolerance and emotion regulation. The methodological approach to theory development is new and interesting. 

1. Abstract: Please correct typos in the last sentence. 

2. Page 2, lines 52-53: „…, it can be central to the destruction of hope, resulting in major psychological consequences such as severe depression or even suicide.“ This sounds a little drastic. Perhaps the authors could tone it down a little. 

3. Page 2, lines 55-62: I was thinking whether it may be conductive to understanding the relationship between hope and goal achievement if the definition by Snyder et al. was introduced at the beginning of the paragraph. 

4. It seems to be closely related to constructs such as self-efficacy and actions vs. state orientation

5. Page 3, lines 75-77: How does uncertainty influence the ability to hope? Because the perceived lack of information complicates the definition of goals which makes it difficult to mobilize goal-directed energy. 

6. Pages 3-4, lines 98- 110: This section is difficult to understand for readers who are not familiar with social science models of theory formation. 

7. Page 5, line 178: Grammar

8. Page 8, line 260: I am not sure whether I missed this piece of information but it did not really become clear to me from Figure 2 and the text how exactly eligibility criteria was expanded. 

9. Figure 5: Why does the history of emotional disorders influence “The Unknown”? What about personality disorders? What does “-ve” and “+ve” mean? I believe that readers could benefit from figure legend introducing the main tenets of the final model. Please introduce the abbreviations. 

10. Page 10, lines 357-358: “…providing superficial support for treatment interventions such as Cognitive Behavioural Therapy (CBT)”. I am a little confused here as mindfulness plays a major role primarily in third wave CBT approaches, such as Acceptance and Commitment Therapy (ACT) and Dialectical Behaviour Therapy (DBT). Mindfulness is not a central component of second wave CBT. Maybe you could be a little more specific here. 

11. Page 10, lines 420-422: The various explanations of uncertainty intolerance are not mutually exclusive. Someone who has a very sensitive Behavioral Inhibition System and inadequate self-soothing skills is less tolerant of unpleasant sensations. Maybe you could think about how the different theoretical conceptualizations of IU relate to each other. 

12. Page 10, lines 428-430: To what extent are interoceptive processes dysregulated in people with mental illness? I think that this aspect is not self-evident and needs clarification. 

13. Page 10-11, lines 434-456: I am missing the common thread in this paragraph. It does not clearly convey how exercise therapy/ graded exposure influences core constructs of the final model and how it helps to restore a sense of identity. The paragraph introduces the complex construct “self-identity” without defining it or without being embedded in a theoretical background and ends by citing research about the impact of CBT on hope and pain intensity. 

14. Page 13, lines 459-460: Does hope influence pain perception or is it the other way around? The study by Wojtyna et al. found that hope is influenced by experiences of pain and pain in the present moment. If the study describes associations rather than causal relationships, please formulate the statements accordingly. 

15. Page 13, lines 469-470: Is CLBP really “poorly” understood? Maybe you could phrase your ideas behind this expression differently. Self-efficacy and self-identity are being mixed up somehow

16. Page 14, lines 501-502: Is cognitive flexibility really a component of emotion regulation or does it result from adaptive implicit and explicit emotion regulation? Others would argue that cognitive rigidity would result from the inability to access brain systems supporting the generation of new self-congruent goals (compare Kästner and Petzke (2024). Personality systems interactions theory: an integrative framework complementing the study of the motivational and volitional dynamics underlying adjustment to chronic pain. Front. Pain Res.). 

17. Page 14, lines 509-510: There is a recent meta-analysis summarizing the evidence from randomized controlled trials investigating the efficacy of Acceptance and Commitment Therapy for chronic pain. Maybe you want to cite this: Ma et al. (2023). The Efficacy of Acceptance and Commitment Therapy for Chronic Pain. A Systematic Review and Meta-analysis. The Clinical Journal of Pain. 

Reviewer 2 Report

Comments and Suggestions for Authors

Introduction

1.       Page 2, line 49, Define what ERS is.

2.       The introduction is unsatisfactory, as it fails to discuss important concepts such as the distinct mechanisms underlying different types of CLBP (nociceptive, neuropathic, and nociplastic pain). The claim that the majority of CLBP cases lack identifiable structural causes is unsupported by data. Moreover, the rationale for focusing on hope and uncertainty over other psychological constructs—such as anxiety, depression, or knowledge deficits—remains unclear. The introduction also provides no explanation of how these factors affect treatment outcomes or recovery. The definitions of hope and uncertainty specific to CLBP are vague; for example, is "hope" related to pain improvement or the identification of a pain cause? Similarly, what does "uncertainty" encompass in this context? Additionally, while emotional regulation appears central to the proposed framework, the authors fail to introduce or systematically review this concept. This omission undermines the theoretical model and creates a confusing and weak rationale.

Methods

1.     The exclusion of participants over 70 years old is unconvincing. The cited reference is outdated from 10 years ago and does not reflect current data. Furthermore, the review does not suggest that CLBP prevalence decreases in this demographic. Instead, it indicates consistent prevalence among those over 65. Excluding older participants diminishes the study’s inclusivity and relevance.

2.     The authors must explicitly describe the inclusion criteria for selecting articles about uncertainty in the methods section. Figure 2 should complement the written content rather than leaving readers to interpret it independently.

3.     The idea generation process described in Figure 2, step 4, appears incomplete and arbitrary. The premise of question #1—What are the key psychosocial factors that people with CLBP commonly report that they are uncertain about?—is unclear. Most patients are uncertain about the causes and recovery of their pain, making the focus on specific psychological factors less intuitive and harder to justify.

4.     The exclusion of optimism from the search strategy requires explanation.

5.     The PRISMA diagrams for both hope and uncertainty raise questions about validity and reliability, with only 6 articles on hope and 3 on uncertainty included out of 225 initially identified. This low inclusion rate raises concerns regarding whether the appropriate search strategy is applied.  

Results

1.     The authors fail to address how sociodemographic factors influence hope, uncertainty, and emotional regulation in CLBP populations. Ignoring these variables weakens the proposed model (Figure 5) and diminishes the depth and applicability of the discussion.

2.     Figure 5 inadequately explains the interplay between hope, uncertainty, and emotional regulation strategies. The relationships among components are unclear, and the model lacks specificity regarding how these factors yield adaptive or maladaptive outcomes. A revised figure with clearer connections and explanations is necessary.

3.     The review does not provide a detailed analysis of the 9 included articles, despite their centrality to the framework. The methodologies, findings, and relevance of these studies should be thoroughly discussed to establish credibility.

4.     Emotional regulation strategies (ERS) are emphasized in the results section, yet ERS was not systematically reviewed. The inclusion of ERS in the theoretical model is unsupported by evidence, further undermining the review’s conclusions.

Discussion

1.     The relationship between hope and uncertainty is not novel, and the authors’ discussion adds little insight. While the importance of these factors in treatment outcomes is noted, the discussion lacks depth regarding their mutual influence and underlying drivers. Key questions—such as whether uncertainty arises from patients’ lack of knowledge about their condition or clinicians’ failure to address these factors—are not explored.

2.     The proposed theoretical model is difficult to interpret and fails to provide practical guidance for clinical application. It overlooks critical components influencing the interplay between hope, uncertainty, treatment outcomes, and recovery in CLBP populations. While the authors call for further research, the model’s limitations and lack of clarity hinder its utility in advancing understanding or practice.

Reviewer 3 Report

Comments and Suggestions for Authors

Abstract

Must specify that the authors used PRISMA methodology.

Introduction

I suggest you start with the full name of CLBP and then continue with the acronyms.

Method

- The authors refer to using an ethnographic methodology but carried out a systematic review based on the PRISMA model. The authors must describe this methodology in detail.

- If the eligibility criteria indicate that the studios use "population", the authors should replace it with "sample".

- The authors should also explain why was limited the search since 2003.

- The exclusion criteria, e.g. excluding grey literature, are not mentioned.

- In the search strategy, the authors mention that did two systematic searches. The authors should justify why did that.

- The authors should also specify whether the inclusion/exclusion criteria were the same for both systematic searches.

- If the second review is an extension of the first, why didn't the authors decide to expose only the second review?

- The authors should also specify the information search formula, and write the keywords and their Boolean operators for each of the systematic reviews.

- The authors must show the PRISMA graph for each review.

- In point 2.5 on Quality, the authors do not express clearly if did or did not do the evaluation.

- Lines 220-256 has a very long text. The authors should write it in a clear, precise, and brief way. Moreover, the authors seem to be reporting results, so it should not be in the Methods section but in the Results section.

- The authors begin the Results section with a proposal of their model but first should report the results found in the search and analysis of articles and then propose the model, thus ensuring a logical and coherent discussion.

The authors refer to the Limitation that the search was carried out by only one of the authors, so the authors did not comply with the requirements of PRISMA.

I consider that major methodological weaknesses prevent an adequate analysis of it.
